# Assessing the Attractive Effects of Floating Artificial Reefs and Combination Reefs on Six Local Marine Species

Chenglong Han [1], Kefeng Liu [2], Toshihisa Kinoshita [3], Biao Guo [2], Yifan Zhao [1], Yuhang Ye [1], Yufei Liu [1,4], Osamu Yamashita [3], Debin Zheng [2], Wenhui Wang [3] and Xueqiang Lu [1,*]

[1] Tianjin Key Laboratory of Environmental Technology for Complex Trans-Media Pollution, Tianjin International Joint Research Center for Environmental Biogeochemical Technology, College of Environmental Science and Engineering, Nankai University, Tianjin 300350, China
[2] Tianjin Fisheries Research Institute, Tianjin 300457, China
[3] TBR Company Limited, Toyokawa 441-0103, Japan
[4] College of Geography and Environmental Science, Tianjin Normal University, Tianjin 300387, China
* Correspondence: luxq@nankai.edu.cn

**Abstract:** Artificial reefs (ARs) have been advocated for and implemented as management tools for recreational fisheries, species conservation, and habitat replacement; however, the research that includes attracting marine species of floating ARs remains in its early stages. Here, two types of floating ARs were designed to evaluate the attractive effects using the occurrence rate and attracting index for six commercially important species (*Lateolabrax maculatus, Liza haematocheila, Sebastes schlegelii, Acanthopagrus schlegelii, Litopenaeus vannamei,* and *Amphioctopus fangsiao*) in the Bohai Bay of China; their combined ARs were meanwhile compared with two variants of artificial seagrass beds (SA and SB) and the traditional double-frame artificial reef (TD). All of the designed ARs were effective in attracting experimental species. The ARs with higher shelter areas (SB and TD) showed a better attracting effect. The efficiency of the ARs in attracting different species depended on their life histories. The bottom-mounted ARs were more efficient in attracting demersal species, while the floating ARs attracted epipelagic fish. In addition, the combined reefs had a better attractive effect than single ARs did. Overall, floating ARs and their combined ARs show the potential to deploy especially for attracting epipelagic fishes, although further study is needed.

**Keywords:** floating artificial reef; tank-scale attracting experiment; effectiveness evaluation

**Key Contribution:** The floating artificial reefs were attractive to epipelagic fishes, and floating artificial reefs, as well as their combinations with bottom reefs, have the potential to reconstruct marine habitats.

## 1. Introduction

Marine fisheries are a critical global food supply element and are significant for many countries and regions [1]; however, many marine species, habitats, and ecosystems have suffered catastrophic declines [2–4] due to increased pollution, the development and hardening of coastlines, the extraction of resources, and the introduction of invasive species [5–9].

Constructing artificial reefs (ARs) is a viable solution for the deteriorating marine species, their habitats, and ecosystems [9,10]. ARs have been used to attract fish from surrounding habitats in addition to serving as a management tool for stock protection and enhancing fishery resources by providing alternative adult habitats and juvenile nursery as well as spawning ground habitats [11–13]; however, the success of the deployment of ARs depended on their physical properties, such as configuration, material composition, shelter, shading offering, and the effectiveness on attracting marine fish [14,15].

In recent decades, millions of dollars have been spent worldwide on designing and building ARs in order to improve their functions [16,17]; however, deployed bottom ARs have better recovery and proliferation effectiveness with regard to demersal fishes, oysters, and sea cucumbers than epipelagic fishes [18,19]. Epipelagic fishes and their habitats may have been neglected in biological restorations using ARs. Floating ARs are a feasible method for building artificial structures floating in the water [20]. They can potentially use marine space and display excellent antiwave as well as current performance, less constraint on seabed geology, and a feasible arrangement mode of multireef groups [21]; however, the research on the use of floating ARs to restore the ecological environment was still in its initial stage. Furthermore, there are gaps in evaluating the materials, configurations, and attraction effectiveness of floating ARs, which requires proper research.

The most common reef material is concrete [22], which has a stable structure, chemical properties, and service durability [10,23]; however, over time, ARs, such as concrete, can become immersed in sediment sand and become ineffective [24]. Additionally, transportation difficulties and the installation of large and heavy ARs hindered the construction of high-rise and floating ARs. Recently, plastic-based materials, such as polyvinyl chloride (PVC) and polypropylene (PP), have shown the potential of constructing high-rise ARs and keeping the reefs elastic with easy installation, lightweight, and with comparable durability [25–27]; however, the studies on ARs using flexible materials and their performance assessment are relatively few.

Successful deployment showed that complex ARs could directly increase the availability of vital resources, such as food and shelter, and mediate biological interactions [28–31]. ARs with more shelter and shading supported higher fish abundance and diversity [32,33]. Studies have demonstrated that certain fish species prefer vertical terrain, and that this preference is correlated with greater fish numbers [34,35]. Floating Ars can be combined with bottom ARs to create the possibility for vertical and complex habitats. Nevertheless, there is no precedent for this, and no research has been carried out on their potential to attract and recover marine fish.

Approximately 50% of AR performance review case studies found evidence of successful outcomes after deployment [36]. Deployments frequently disregard the specific habitat requirements of local marine species and the degree to which they consider such areas viable habitats [36]. Before deploying ARs, selecting local marine fishes with which to conduct ecological attracting experiments is necessary to achieve the expected ecological and economic advantages; however, the present experiment's choice of experimental species and AR types is limited and deficient in systematics and sustainability [37]. Additionally, it is unclear how to configure the artificial reef types and combinations for different target organisms in order to achieve the best results. To provide guidelines for deploying and researching ARs, especially floating ARs, it is necessary to mimic the natural environmental conditions in the laboratory and perform systematic, qualitative, and quantitative research on multispecies and multi-AR ecological attraction.

In this study, we developed two types of floating ARs and two types of bottom ARs through the use of PP materials. The performance of the floating ARs and combined ARs was evaluated through the use of six typical economic species in the Bohai Bay. The objective was to determine how different AR configurations, especially floating ARs, affect the attracting effect of marine species.

## 2. Materials and Methods

### 2.1. Experiments

Experiments were conducted in an indoor aquaculture tank from July to September 2021. The pond measured 6 m by 4 m by 1.5 m and was coated in cement as well as weatherproof paint. The temperature was set at 25 °C, and the water depth was kept at 1.0 and 1.2 m.

We selected 6 representative economic species (Table 1) in Bohai Bay according to fishery data from the Ministry of Agriculture and Rural Affairs of China. These species included two epipelagic fishes *(Lateolabrax maculatus* and *Liza haematocheila)*, two demersal

fishes (*Sebastes schlegelii* and *Acanthopagrus schlegelii*), *Litopenaeus vannamei*, and *Amphioctopus fangsiao*. *L. vannamei* was a substitute for *Penaeus orientalis*, which was scarce in the Bohai Sea and difficult to keep alive after catching, and was the same family (*Penaeidae*) with a similar appearance and habits. The Tianjin Fishery Research Institute provided all of the experimental species. Prior to the experiment, the species were temporarily raised in tanks for at least 72 h. The reefs were also placed into the tank to adapt to the environment and avoid stress; each time, 30 healthy species were used in the experiment.

**Table 1.** Characteristics of experimental species (*n* = 30).

| Species | Age | Length (cm) | Weight (g) | Source | Note |
|---|---|---|---|---|---|
| *Lateolabrax maculatus* | Juvenile | 12.56 ± 0.78 | 94.45 ± 5.35 | | |
| *Liza haematocheila* | Juvenile | 4.26 ± 0.77 | 1.56 ± 1.47 | | Epipelagic fishes |
| *Sebastes schlegelii* | Juvenile | 6.26 ± 0.84 | 14.18 ± 3.26 | Breed aquatics | |
| *Acanthopagrus schlegelii* | Juvenile | 5.37 ± 0.32 | 8.54 ± 0.85 | | |
| *Litopenaeus vannamei* | Juvenile | 12.88 ± 2.24 | 11.49 ± 1.58 | | Demersal species |
| *Amphioctopus fangsiao* | Juvenile | 18.38 ± 4.26 | 157.36 ± 32.70 | Netted | |

Four ARs (Figure 1a), i.e., floating AR Type A (FA), floating AR Type B (FB), artificial seagrass bed Type A (SA), and artificial seagrass bed Type B (SB), were designed and assembled via the use of PP woven ropes. Buoys were installed to cause the ARs to float straight. FA was a semi-closed hollow square structure surrounded by artificial seaweed. FB was an open square structure arranged with artificial seaweed in parallel. SA and SB had the same amount of artificial seaweeds, but SB was arranged more closely. In addition, one traditional double-layer frame reef (TD), made from high-density PVC, was used as the common AR for comparison.

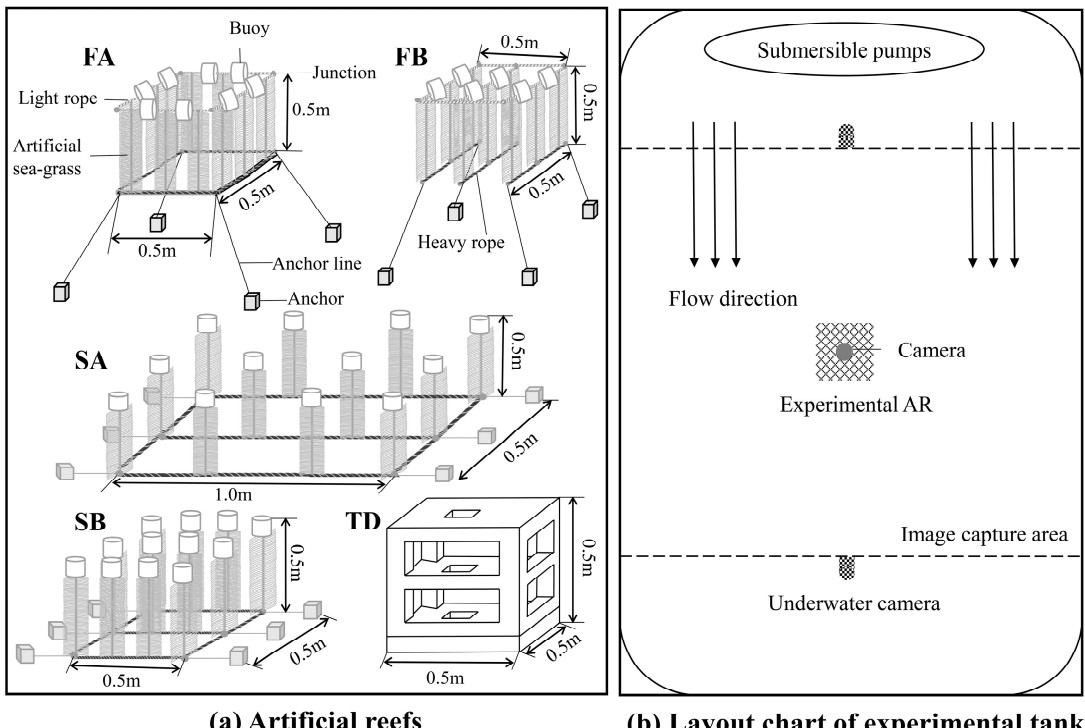

**(a) Artificial reefs**　　　　**(b) Layout chart of experimental tank**

**Figure 1.** Configurations of artificial reefs (ARs) and the layout of the experimental tank. (**a**) ARs: FA (floating artificial reef Type A), FB (floating artificial reef Type B), SA (artificial seagrass bed Type A), SB (artificial seagrass bed Type B), and TD (traditional double reef). (**b**) Layout chart of the experimental tank. The area surrounded by dotted lines was the image capture area.

Single-AR tests for the FA, FB, SA, SB, and TD (Experiment I), as well as combined AR experiments for six combined ARs (FA + TD, FB + TD, FA + SA, FA + SB, and FB + SB) (Experiment II), were performed in two separate groups (Figure 2). The control without any ARs was set under the same conditions. Six species were designed into the attraction experiments for each AR model; however, only one species and one AR model were involved in each tank, rather than putting all six species into a tank, as there could be an invisible predatory relationship. A total of 72 groups of experiments on 6 experimental organisms were conducted, including 30 groups in Experiment I, 36 groups in Experiment II, and 6 groups of control experiments. At the beginning of the experiments, the AR model was placed in the center of each tank, and the experimental species (30 tails) were then placed. Cameras were used to record the behavior of the organisms. There was no feeding or aeration during the day, and these were conducted at night. After each group of experiments, the experimental water, AR models, and experimental individuals were replaced.

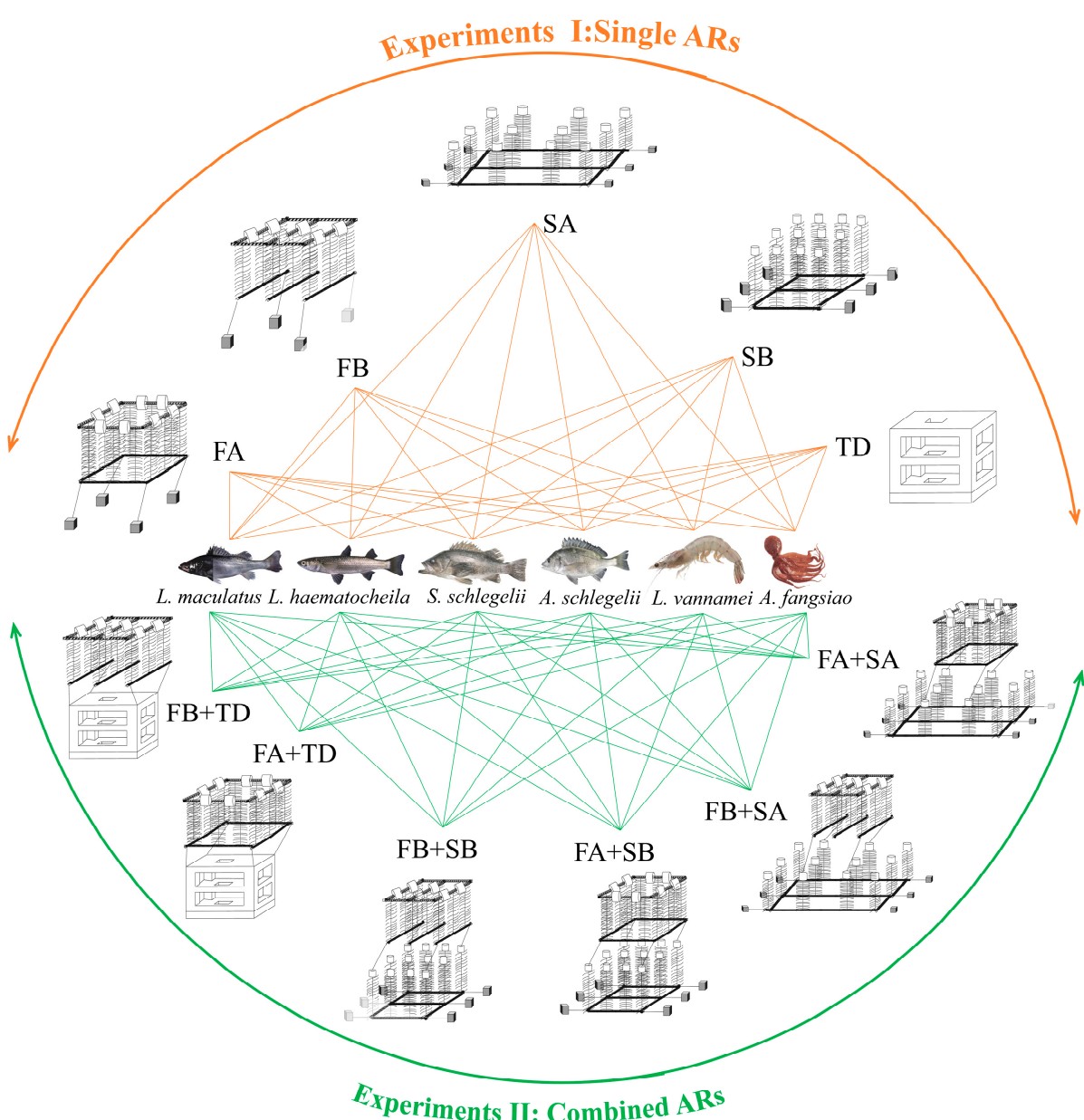

**Figure 2.** Attracting experiments. Each connecting line represents a group of experiments. The ARs included FA (floating artificial reef Type A), FB (floating artificial reef Type B), SA (artificial seagrass bed Type A), SB (artificial seagrass bed Type B), and TD (traditional double reef).

The experimental tank layout is shown in Figure 1b. Three submersible pumps were installed to generate water flow to simulate the ocean current. The designed flow velocity was set as the average flow velocity of $35.7 \pm 0.49$ m/s in the Bohai Bay [38]. High-definition underwater cameras (GW0108CD, Shenzhen Meiyijia Electronic Technology Co., Ltd. of China, Shenzhen, China) were used to observe and record species distribution during the experiment. One high-definition camera was set at 1.5 m above the water surface, and two high-definition underwater cameras were set at 1.5 m from the reef on the opposite side. The images recorded by the cameras were transmitted through a cable to the recording machine. A computer was used for real-time observation and recording in the monitoring room. The recording time for each experiment was 1.5 days, and the image capture time was from 9:00 to 11:00 and from 14:00 to 16:00 daily. Each experiment evaluated the attracting effect based on the 6 h of video. The experiments were conducted under natural light, but LED lamps ensured light intensity.

*2.2. Data Analysis*

The evaluation of the attracting effect of ARs was based on the 6 h of images segmented by an interval of 1 min through the use of MATLAB. A total of 360 images were captured for each experiment. The number of organisms in the ARs in each image (*n*) was recorded.

The average occurrence rate (*R*, %) was defined as follows:

$$R = \frac{\sum_{i=1}^{m} n_i}{mk} \times 100 \tag{1}$$

where $n_i$ is the occurrence number of species in the reef area for image *i*, *m* is the total captured image number ($m = 360$), and *k* is the experimental species number ($k = 30$).

The attracting efficiency index (*I*) was also defined as the ratio of the average occurrence rate (*R*) of the experimental species to the volume (*V*) of the experimental ARs [39]:

$$I = R/V \tag{2}$$

where *I* is the attracting efficiency index, *R* is the average occurrence rate, and *V* is the volume of the experimental ARs. The larger *I* is, the better the attractive effect of ARs.

All of the data were expressed as mean $\pm$ standard deviation, and SPSS 20.0 was used to compare species and AR configuration differences via an ANOVA. The Bonferroni method was used for pairwise comparison to correct *p*-values; $p < 0.05$ was considered statistically significant.

## 3. Results

*3.1. Attracting Effect of Single ARs*

Figure 3a shows the attracting efficiency index *(I)* for different single ARs (Table A1). Generally, the attracting effect of AR types on species showed significant differences ($p < 0.05$). All of the tested ARs showed attractive effects compared to the control group, and the effects varied based on the ARs and species (Figure 3a, Table A1). Most ARs effectively attracted *L. maculatus* and *A. fangsiao* (Figure 3a, Table A1); however, most ARs had marginally attractive effects on *A. schlegelii* and *L. vanamei* (Figure 3a, Table A1). FB and SB had the best attracting effects for *L. haematocheila* and *S. schlegelii*, respectively (Figure 3a, Table A1). SB and TD had relatively good attracting effects for most species (Figure 3b).

*3.2. Attracting Effect of Combined ARs*

As shown in Figure 3c, all of the combined ARs showed a higher attracting efficiency index *(I)* than the control (Table A1). The attracting effect of combined AR types on species showed significant differences *(p < 0.05)*. The different attractive properties of the combined ARs varied according to the species and kind of reef (Figure 3c, Table A1). For the majority of species, FB + SB demonstrated acceptable attractive properties (Figure 3c, Table A1). The *I* value of FB + TD for *L. maculatus* was the highest (Figure 3c, Table A1). FB + SB

had the best attracting effects for *L. haematocheila*, *S. schlegelii*, and *A. schlegelii* (Figure 3c, Table A1). FA + SA and FA + TD had the best attracting effects for *L. vannamei* and *A. fangsiao*, respectively (Figure 3c, Table A1).

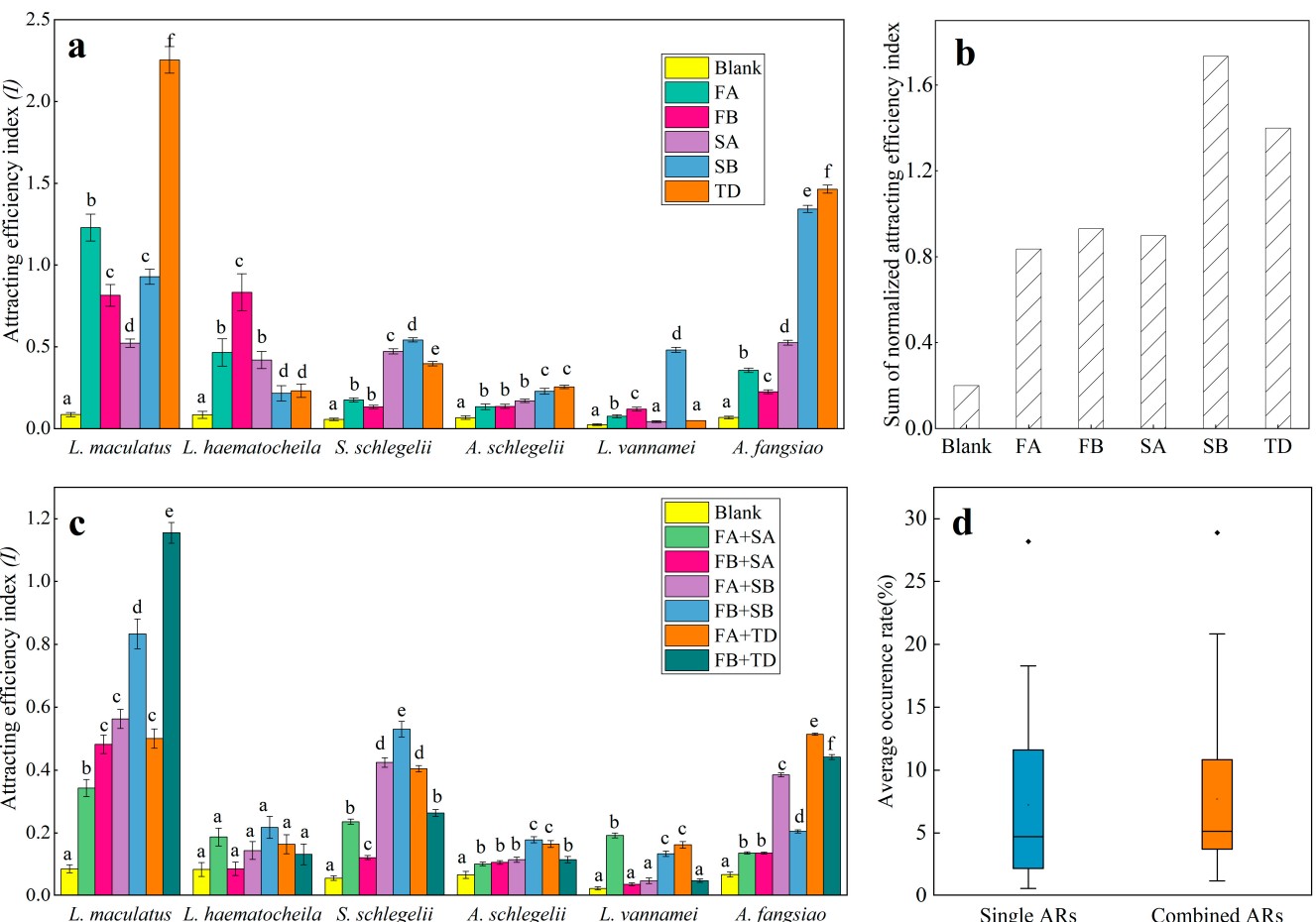

**Figure 3.** (**a**) Attracting efficiency index *(I)* of five artificial reefs (ARs). The ARs include FA (floating artificial reef Type A), FB (floating artificial reef Type B), SA (artificial seagrass bed Type A), SB (artificial seagrass bed Type B), and TD (traditional double reef). Blank was the control. (**b**) The attracting effect of each AR, the vertical coordinate, was the sum of *I* standardized. (**c**) Attracting efficiency index of six combined ARs. (**d**) Average occurrence rate of test species on single ARs and combined ARs. Lowercase letters had the same meaning as (**a**). For the letters on each data bar, if no same letter can be found on two data bars, the difference between the two data is statistically significant ($p < 0.05$); otherwise, the difference between two compared data is not statically significant. *p*-values were corrected by the Bonferroni method.

## 4. Discussion

### 4.1. Influence of Species Ecology

The design of ARs should consider the targeted species' biological needs, such as offering shelter to adults, eggs, or larvae [40]. The observed differences in attracting different ARs (Figure 3a) might be related to the ecological habits of the species. The experimental species' community ecology may have influenced their behavior of clustering or dispersing into the tested reef.

The target species' life cycle was essential for deploying ARs. *L. haematocheila* is a eurytherm and euryhaline fish that prefers to be scattered in shallow coastal waters [41,42]. Because of their photosensitivity, their young fish avoid reef settings. Hence, the floating ARs attracted *L. haematocheila* better than other reefs did. Some demersal fishes tend to move in a large group. *S. schlegelii* and *A. schlegelii* prefer rocky sea habitats and dwell in

concealed as well as shadowy places. *A. fangsiao* uses coral reefs and rocks as burrowing and hiding habits [43]. This being the case, they had a higher I value in SB and TD than other species. *L. vannamei*'s natural habitat is muddy seafloor, and they are nocturnal. Except for SB, the ARs showed low attractivity for *L. vannamei* in light-exposed experimental conditions. In addition, individual body sizes influence the attractive effect. The I value of *L. maculatus* was significantly higher among the four fishes. The value might be explained by *L. maculatus* individuals being larger than other fishes, their gonads tending to be larger when they first mature, and the reef's strong ability to draw in huge fishes [44].

*4.2. Influence of ARs' Configuration*

The ARs' configuration is the key factor in influencing the attraction effect. Studies have shown that artificial materials' microstructure complexity and durability can strongly influence attraction [3]. More complex structures attracted more species [45,46]. Five kinds of individual ARs were put into the laboratory tank. Their attraction to the tested organisms mainly depended on the reef's internal space area and shadow area. In this study, SB and TD had the best effectiveness in the single-AR attracting experiment compared to other reef types. The attraction was because SB had a large specific surface area and a close arrangement of artificial seagrass, which provided good coverage and a more shaded area. TD was open on both sides and had a roof structure, allowing species to shuttle and gather (Figure 3b); however, the attractive effect of SA was lower than that of SB and TD. The sparse arrangement of artificial seagrass in SA produced less concealed space with a worse covering effectiveness than in SB, although the number of artificial seagrasses in SA and SB was the same.

FA and FB were less effective than the other reef types and they had similar attractive effects on the test species (Figure 3a,b), most likely a result of the fact that FA and FB did not offer as much shadow area as SB and TD did and were situated in the upper water body with high light intensity. Even so, they had better attractive effects on epipelagic fish, such as *L. haematocheila*, but poorer attractive effects on species with negative phototropism, such as *S. schlegelii*. Due to the limited effective space and depth of water, the floating ARs may be unable to exert their full function and be insufficient in providing effective shelter for demersal fishes. The limitations of water depth and pool space not only affected the effectiveness of the artificial reef we designed but may also have an impact on the organisms. For example, the shallow water depth may have an impact on the behavior of benthic fish, shrimp, and *A. fangsiao* adapted to the dim and dark environment of the bottom, driving them to seek other shelter or gather around the tank. In addition, the attracting effect of FA was weaker than that of FB (Figure 3a,b), which may be concerned with their structural differences. FA was a semi-enclosed structure that surrounded artificial seaweeds on the side and had a hollow interior, making it impossible for organisms entering the FA to find shelter inside. FB was open on both sides to facilitate the swimming of organisms, and there were artificial seaweeds that provided shelter inside.

Our results indicated that the attractive effects of AR types on the tested species were significantly different ($p < 0.05$); however, pairwise comparisons using the Bonferroni method showed that not all configurations differed significantly in their attracting effect (Figure 3a). In other words, there was no significant difference between the false-positive results based on the degree of similarity between specific AR configurations and the attractive power of the studied species. For instance, all bottom reefs attracted *L. haematocheila* similarly, while SB and TD had the same impact on *S. schlegelii*.

Nevertheless, the significant differences ($p < 0.05$) between the five ARs in attracting experimental species reflected the differences in species habitat selection. Five ARs simulated two different habitat types. The seagrass beds (SA and SB) and the traditional reef (TD) indicated the bottom layer of the ocean substitute for habitat. In contrast, the floating reefs (FA and FB) represented the middle and upper layers. Only the bottom ARs were studied and included in the comparisons with other research (Figure 4); however, the results reveal that while bottom ARs had better effectiveness in attracting other species, floating ARs had

superior effectiveness for attracting epipelagic fish. The attraction might be a combination of life history and configuration; the experimental species were inclined to choose habitats that met their life history habits. Given the greater availability of shelter space, complex seabed structures with small niches may be more attractive than floating reefs for species that prefer reef habitats [47,48]. The floating reefs may provide a middle-to-upper level of refuge and food for epipelagic fish [49].

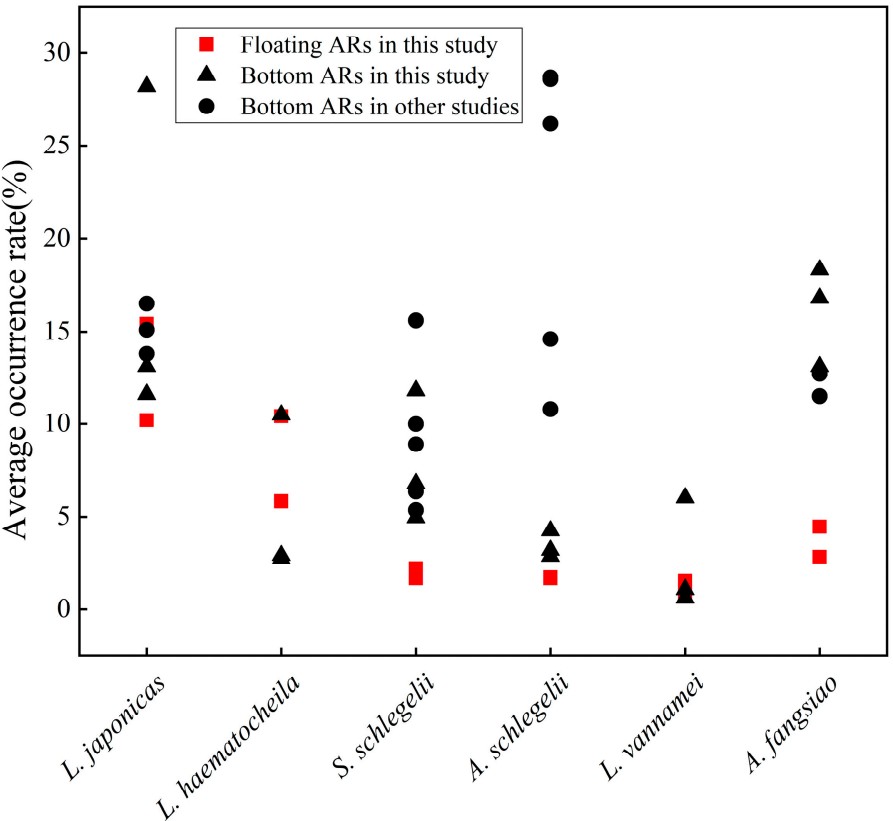

**Figure 4.** The average occurrence rate of experimental species on floating artificial reefs (ARs) and bottom ARs, in addition to a comparison with other studies.

*4.3. Would the Combined ARs Be Better?*

We compared the attracting effects of various combinations of floating ARs and bottom ARs on experimental species. Except for *L. haematocheila*, the attractive effects of all configurations had significant differences ($p < 0.05$), demonstrating that various vertical assemblage structures had different attractive effects on the tested species. In totality, FA + SA and FB + SA had fewer attracting effects on species, which may be related to the sparse arrangement of artificial seaweed in SA. It produced less hidden space and covering effectiveness, resulting in weaker attracting effects of these compared to other reef types, which was the same as the conclusion obtained above. Additionally, FB + SB and FB + TD had a better attracting effect on the tested species than other combination modes did (Figure 3c), the same as the attracting effect of single ARs above. FB was better than FA, while SB and TD had the best single-reef attracting effect. Naturally, these two combinations, supporting more shelter and space, had the best attracting effects. Pairwise comparisons using the Bonferroni method (Figure 3c) revealed that all of the combinations on *L. haematocheila* generated poor and insignificant attractive results. This is probably because of their propensity to underuse vertical combinations because juvenile fish have a light tropism characteristic.

Complex and sheltered ARs can attract more marine organisms. We examined the attractive effects of combined and single ARs (Figure 3d). The combined ARs were marginally

better (7.71%, $N = 36$) than the single ARs (7.25%, $N = 30$), but there was no difference in the Wilcoxon signed-rank test results ($Z = -0.586$, $p = 0.558 > 0.05$). As a result, although similar to the single ARs, the combined ARs showed the desired attracting effect. The complex and integrated pulling power of ARs relied on how effectively the habitat served as a place for species to find food, shelter, and spawning grounds [50,51]. We conjectured that the size of the tank and the combined AR model might be limited. The tank had attained saturation with the 30 species added when the ARs were the specified size. Other juveniles who could not access the reef area collected in the shadow region outside the reef area due to the territorial behavior displayed by juveniles in the reef area to drive away other juveniles who attempted to enter their territory. In addition, species followed their life histories. The epipelagic fishes cautiously visited the upper habitats, and the bottom fishes cautiously caught the upper habitats.

Our comparison indicated that the combination of floating ARs and bottom ARs may be more attractive to marine species. Coupled floating ARs may be utilized as a workaround for attracting different species in the ocean, since they can ensure integrated habitats for demersal and phototactic species; however, the use of floating ARs and their combinations in ecological restoration is still in its infancy. The design of floating ARs originated from fish aggregation devices (FADs), which have been used for many years [52]; however, FADs are used to aggregate fish quickly to make it simpler for fishermen to catch them. In contrast to FADs, floating artificial reefs should be monitored over a larger region and for a longer length of time to benefit fish populations and improve productivity. Additionally, many coastal countries in the world have constructed marine pastures and placed a large number of ARs, of which the vast majority were bottom ARs. Due to their limited height, the impact on the waters above the ARs may be relatively insufficient, and ARs were unable to fully utilize the entire water space. In order to improve comprehensive and three-dimensional development, it may be a considerable practice to combine bottom ARs in conjunction with floating ARs and jointly release them; it has been established that the combination of floating and bottom reefs complimented one another and impacted comprehensive usage [21].

Building the combined ARs was effective, and the combination configuration, which offers additional refuge capacity, may enhance species abundance, as proven in previous studies. We studied the attractive effect of ARs on six species in tanks, but the results may not be the same as those of ARs placed in the sea, which tended to have a higher abundance of species. ARs were not limited to providing shelter but also reasonable feeding and spawning grounds. Fish ecology in reef fisheries showed that fish gathering was closely related to baiting and escape behavior [53]. It is primarily bait density, possible exit points, and the need for bait that influence how species behave around the reef [54]. Additionally, the composition of benthic communities that settle on artificial reefs can strongly influence reef attraction [3]. Due to the significant difference in the artificial bait from the species' actual diet and the fact that casting bait would not have the desired attracting effect in the sea, this experiment did not cast bait in the reef area. Moreover, by boosting spawning success and early life stage survival rates, the creation of ARs is anticipated to promote the objective of species recovery [40]. According to studies, artificial reefs can serve as spawning enhancement tools by enticing adult fish to spawn and creating favorable conditions for successful egg deposition, hatching, and larval emergence [50,55]. In the future, we will focus on integrating novel flexible ARs and traditional reefs into the water, observing and collecting samples of the eggs and organisms around, inside, and attached to ARs using diving observation, underwater photography, fishing, and other techniques with which to investigate the actual attracting of ARs and provide a reference for the design and placement of ARs as well as the development of marine ranching.

## 5. Conclusions

In this study, all of the designed ARs effectively attracted the six experimental species. The ARs with higher shelter areas showed a better attracting effect. The effectiveness of the

ARs in attracting different species depended on their life histories. The bottom ARs were more effective for the demersal species, and the floating ARs were effective for epipelagic fishes. In addition, for the six experimental species, the attracting effect of the combined floating ARs was slightly better than that of the single ARs. This study could not evaluate other attractive effects, such as hunting and egging, because of the time and size limitations. The minimal impact, including the shelter effectiveness, can be said to be attractive. In general, floating ARs and their combined ARs have the potential to use maritime space better, although this area still needs further study.

**Author Contributions:** Conceptualization, methodology, writing, and visualization, C.H. and X.L.; formal analysis and data curation, C.H.; investigation, C.H., Y.Z., Y.Y. and Y.L.; resources, K.L., B.G., D.Z., T.K., O.Y. and W.W.; supervision, project administration, and funding acquisition, X.L. All authors have read and agreed to the published version of the manuscript.

**Funding:** This research was funded by the National Key R&D Program of China, grant number 2019YFE0122300.

**Institutional Review Board Statement:** The animal study protocol was approved by the Animal Experimental Ethical Committee of the Laboratory Animal Centre, Nankai University. Approval code: 2023-SYDWLL-000509.

**Data Availability Statement:** All data generated during this study are included in this published article.

**Acknowledgments:** We thank Weifu Zhang and Xian Song from Tianjin Xingsheng Aquaculture Co., Ltd. for their care of experimental species.

**Conflicts of Interest:** The authors declare no conflict of interest.

**Appendix A**

**Table A1.** Attracting efficiency index of five artificial reefs (ARs) and six combined ARs. The ARs include FA (floating artificial reef Type A), FB (floating artificial reef Type B), SA (artificial seagrass bed Type A), SB (artificial seagrass bed Type B), and TD (traditional double reef). Blank was the control. All data were expressed as mean $\pm$ standard deviation ($n$ = 360).

| Species / ARs | *L. maculatus* | *L. haematocheila* | *S. schlegelii* | *A. schlegelii* | *L. vannamei* | *A. fangsiao* |
|---|---|---|---|---|---|---|
| Blank | 0.086 ± 0.013 | 0.084 ± 0.022 | 0.057 ± 0.007 | 0.067 ± 0.011 | 0.024 ± 0.005 | 0.068 ± 0.008 |
| FA | 1.229 ± 0.082 | 0.466 ± 0.084 | 0.176 ± 0.012 | 0.134 ± 0.017 | 0.076 ± 0.010 | 0.357 ± 0.014 |
| FB | 0.815 ± 0.066 | 0.834 ± 0.110 | 0.133 ± 0.010 | 0.136 ± 0.012 | 0.120 ± 0.012 | 0.226 ± 0.012 |
| SA | 0.523 ± 0.026 | 0.420 ± 0.052 | 0.472 ± 0.016 | 0.170 ± 0.010 | 0.043 ± 0.005 | 0.526 ± 0.015 |
| SB | 0.928 ± 0.046 | 0.216 ± 0.047 | 0.542 ± 0.013 | 0.228 ± 0.018 | 0.481 ± 0.016 | 1.343 ± 0.021 |
| TD | 2.254 ± 0.081 | 0.232 ± 0.041 | 0.397 ± 0.014 | 0.256 ± 0.010 | 0.049 ± 0.007 | 1.464 ± 0.025 |
| FA + SA | 0.342 ± 0.027 | 0.187 ± 0.028 | 0.238 ± 0.008 | 0.103 ± 0.006 | 0.189 ± 0.007 | 0.138 ± 0.003 |
| FB + SA | 0.478 ± 0.029 | 0.086 ± 0.021 | 0.120 ± 0.007 | 0.108 ± 0.006 | 0.037 ± 0.005 | 0.140 ± 0.003 |
| FA + SB | 0.558 ± 0.026 | 0.137 ± 0.028 | 0.417 ± 0.015 | 0.110 ± 0.008 | 0.048 ± 0.009 | 0.376 ± 0.006 |
| FB + SB | 0.827 ± 0.047 | 0.220 ± 0.035 | 0.528 ± 0.025 | 0.180 ± 0.010 | 0.129 ± 0.008 | 0.211 ± 0.005 |
| FA + TD | 0.500 ± 0.031 | 0.157 ± 0.030 | 0.398 ± 0.010 | 0.160 ± 0.011 | 0.157 ± 0.010 | 0.512 ± 0.003 |
| FB + TD | 1.154 ± 0.034 | 0.133 ± 0.033 | 0.264 ± 0.011 | 0.116 ± 0.010 | 0.049 ± 0.005 | 0.442 ± 0.007 |

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
