# Peer review of "Assessing the Attractive Effects of Floating Artificial Reefs and Combination Reefs on Six Local Marine Species"

_fishes, doi:10.3390/fishes8050248_

Round 1
Reviewer 1 Report
This is a well-described work with good figures. The design of the artificial reefs could be duplicated successfully by readers. The writers have provided extensive citations.
Are you sure that your shrimp was Litopenaeus vannamei, which is native to the eastern Pacific and not reported from China? Or did you obtain cultured L. vannamei? In a study in a laboratory tank, there should not be a difference between behavior of species from your area and non-native species. In a natural environment, non-native species may behave differently from local species.
Author Response
Dear Reviewer,
We would like to thank you for your careful reading, helpful comments, and constructive suggestions, which have significantly improved the presentation of our manuscript. We hope that our work can be improved again. Furthermore, we would like to show the details of the revisions, as follows:
General comment:
This is a well-described work with good figures. The design of the artificial reefs could be duplicated successfully by readers. The writers have provided extensive citations.
>> Many thanks for the useful comments. We have revised the manuscript carefully following the comments. The response was made to comments, and we also specified the line numbers in the revised manuscript, which is helpful to find out how we changed.
Comments and Suggestions for Authors:
Are you sure that your shrimp was Litopenaeus vannamei, which is native to the eastern Pacific and not reported from China? Or did you obtain cultured L. vannamei? In a study in a laboratory tank, there should not be a difference between behavior of species from your area and non-native species. In a natural environment, non-native species may behave differently from local species.
Response: Thanks for your comments. The species of shrimp used in this study was Litopenaeus vannamei, which isn’t native species in Bohai Bay. The L. vannamei used in the experiment was artificially bred and raised by local aquaculture plants, which are their main species. Actually, Penaeus orientalis, a native shrimp species of Bohai Bay, is the target species. However, it is difficult to capture sufficient numbers (at least 300) of wild P. orientalis for our experiments due to the scarcity of wild P. orientalis in Bohai Bay. Moreover, it is also difficult to keep wild P. orientalis surviving after capture. Therefore, we used another alternative species (L. vannamei) with similar habits to P. orientalis, which should also be easily available. L. vannamei is in the same family as P. orientalis and they have similar appearance and habits. We have added the above description in the manuscript (Lines 98-100). It is worth noting that we caught a considerable amount of L. vannamei during the ecological survey in Bohai Bay, which illustrated the adaptability of L. vannamei to the Bohai Bay environment, although it may be fled from the local farming plants.
We would like to thank you again for taking the time to review our manuscript and provide your valuable comments. We look forward to receiving your further reply.
Yours sincerely,
Chenglong Han
22 April 2023
Nankai University

Reviewer 2 Report
I have carefully reviewed the article titled "Assessing the attractive effect of floating artificial reefs and its combination reefs on six local marine species" submitted by the authors Han et al. This study provides valuable biological and engineering insights for the development of marine fisheries facilities, such as artificial reefs. Additionally, the research methods, results, and conclusions presented in this paper are generally logical and well-structured. There have been many studies on the correlation between solid-fluid interactions and marine ecosystems, including underwater structures and natural reefs. However, due to various challenges, only a few studies have directly observed these interactions. This paper is significant as it conducted a hydraulic model experiment on real marine organisms and quantitatively obtained experimental results.
The focus of the text is mainly on explaining the structural aspects, such as shade areas, shadows, seaweed density, surface area, and shelter, that are influenced by the shape of artificial reefs. If possible, presenting experimental results from a hydrodynamic perspective would greatly assist researchers in designing marine facilities, such as artificial reefs. For instance, information on the spatial distribution of species and fish could provide insights into the fluid behaviors preferred by fish.
Additional Comments:
1. Is the behavior of the fish constrained by the limited space in the small indoor aquaculture tank used in the experiment?
2. The water depth of the tank used in the experiment was very shallow, and under these conditions, the preference of artificial reefs for fish may differ from that in the actual marine environment. It is necessary to describe the limitations of laboratory conditions in relation to fish ecology.
3. Do the behavioral characteristics and preferences of marine organisms revealed in the experimental results match well with the known research results? If so, the cause needs to be described in detail in connection with the structural characteristics of artificial reefs.
4. Currently, most artificial reefs are installed on the seabed, but why was a floating artificial reef selected as a research subject? The floating type is difficult to apply to in-situ, and the mooring system is difficult to stably maintain its function in extreme environments such as typhoons.

Author Response
Dear Reviewer,
We would like to thank you for your careful reading, helpful comments, and constructive suggestions, which have significantly improved the presentation of our manuscript. We hope that our work can be improved again. Furthermore, we would like to show the details of the revisions, as follows:
General comment:
I have carefully reviewed the article titled "Assessing the attractive effect of floating artificial reefs and its combination reefs on six local marine species" submitted by the authors Han et al. This study provides valuable biological and engineering insights for the development of marine fisheries facilities, such as artificial reefs. Additionally, the research methods, results, and conclusions presented in this paper are generally logical and well-structured. There have been many studies on the correlation between solid-fluid interactions and marine ecosystems, including underwater structures and natural reefs. However, due to various challenges, only a few studies have directly observed these interactions. This paper is significant as it conducted a hydraulic model experiment on real marine organisms and quantitatively obtained experimental results.
>> Many thanks for the useful comments. We have revised the manuscript carefully following each comment. The response was made to each comment, and we also specified the line numbers in the revised manuscript, which is helpful to find out how we changed.
Comments and Suggestions for Authors:
(1) The focus of the text is mainly on explaining the structural aspects, such as shade areas, shadows, seaweed density, surface area, and shelter, that are influenced by the shape of artificial reefs. If possible, presenting experimental results from a hydrodynamic perspective would greatly assist researchers in designing marine facilities, such as artificial reefs. For instance, information on the spatial distribution of species and fish could provide insights into the fluid behaviors preferred by fish.
Response: We agree with the comments. Truly, it would be more interesting to correlate the hydrodynamics with the spatial distribution of tested species, as the reviewer suggested. Regrettably, due to the limitation of the experimental design, we didn’t measure the flow data. Nevertheless, the reviewer suggested a very meaningful point and we would like to test the relationship between fluid changes in artificial reefs and the spatial distribution of species and fish in the next study.
(2) Is the behavior of the fish constrained by the limited space in the small indoor aquaculture tank used in the experiment?
Response: We agree with the comments. We added the discussion (Lines 236-238).
(3) The water depth of the tank used in the experiment was very shallow, and under these conditions, the preference of artificial reefs for fish may differ from that in the actual marine environment. It is necessary to describe the limitations of laboratory conditions in relation to fish ecology.
Response: Thanks for your comment. We have added the description of limitation of the experiment (Lines 238-242).
(4) Do the behavioral characteristics and preferences of marine organisms revealed in the experimental results match well with the known research results? If so, the cause needs to be described in detail in connection with the structural characteristics of artificial reefs.
Response: Thanks for your comment. The connection between the structural characteristics of artificial reefs and the attraction of test organisms has been explained in the manuscript (Lines 218-247).
(5) Currently, most artificial reefs are installed on the seabed, but why was a floating artificial reef selected as a research subject? The floating type is difficult to apply to in-situ, and the mooring system is difficult to stably maintain its function in extreme environments such as typhoons.
Response: The floating artificial reef was chosen as a study subject in order to explore a potential artificial reef to reconstruct the upper habitat that is often ignored. Of course, there are some disadvantages to the floating reef as mentioned by the reviewer. Some advantages are also obvious, including the potential utilization of ocean space and less constraint on seabed geology. In addition, we will combine the floating artificial reefs with seabed artificial reefs and the floating reefs can be moored with the seabed reefs. It may be regarded as the supplement to the seabed reefs.
We would like to thank you again for taking the time to review our manuscript and provide your valuable comments. We look forward to receiving your further reply.
Yours sincerely,
Chenglong Han
22 April 2023
Nankai University

Reviewer 3 Report
The project has been conducted well and the paper should be of wide interest to readers.
One or two features are lacking. I appreciate that the design of floating artificial reefs is in its infancy, and that therefore the experiments have been conducted on small scales in both space and time. This carries a risk of confusion between artificial reefs (ARs, which contribute to productivity of fish populations) and fish aggregating devices (FADs, which simply attract any fish that happen to be in the area, make them easier for fishers to catch, and have no claimed benefits to the fish populations).
Floating FADs have been widely used for many years, and I suggest that the authors should review them, while making clear that the long-term goal of their work is much more ambitious, targeting productivity of fish populations, which FADs do not address. The authors could also make clear that their work is a starting point for design of floating ARs, and that more meaningful implementations would involve bigger ARs that are monitored over much longer time scales. These would be required to test whether ARs benefit fish populations rather than just making fish easier to catch.
The English is good, although has some incorrect usages. For example, the word "effectiveness" should be used in lines 43 and 53 instead of "effect", and "precedent" should be used in line 70 instead of "precedence".
Author Response
Dear Reviewer,
We would like to thank you for your careful reading, helpful comments, and constructive suggestions, which have significantly improved the presentation of our manuscript. We hope that our work can be improved again. Furthermore, we would like to show the details of the revisions, as follows:
General comment:
The project has been conducted well and the paper should be of wide interest to readers.
>> Many thanks for the useful comments. We have revised the manuscript carefully following each comment. The response was made to each comment, and we also specified the line numbers in the revised manuscript, which is helpful to find out how we changed.
Comments and Suggestions for Authors:
(1) One or two features are lacking. I appreciate that the design of floating artificial reefs is in its infancy, and that therefore the experiments have been conducted on small scales in both space and time. This carries a risk of confusion between artificial reefs (ARs, which contribute to productivity of fish populations) and fish aggregating devices (FADs, which simply attract any fish that happen to be in the area, make them easier for fishers to catch, and have no claimed benefits to the fish populations). Floating FADs have been widely used for many years, and I suggest that the authors should review them, while making clear that the long-term goal of their work is much more ambitious, targeting productivity of fish populations, which FADs do not address. The authors could also make clear that their work is a starting point for design of floating ARs, and that more meaningful implementations would involve bigger ARs that are monitored over much longer time scales. These would be required to test whether ARs benefit fish populations rather than just making fish easier to catch.
Response: Thank you for your recognition and valuable comments. Yes, our study is just a preliminary kick-off experiment. We have added the importance of long-term in-situ monitoring of floating ARs effect (Lines 305-310).
(2) The English is good, although has some incorrect usages. For example, the word "effectiveness" should be used in lines 43 and 53 instead of "effect", and "precedent" should be used in line 70 instead of "precedence".
Response: The words have been corrected.
We would like to thank you again for taking the time to review our manuscript and provide your valuable comments. We look forward to receiving your further reply.
Yours sincerely,
Chenglong Han
22 April 2023
Nankai University
